# Donor-Derived Cell-Free DNA as a Non-Invasive Biomarker for Graft Rejection in Kidney Transplant Recipients: A Prospective Study among the Indian Population

**DOI:** 10.3390/diagnostics13233540

**Published:** 2023-11-27

**Authors:** Naveen Kumar, Archita Tandon, Rashmi Rana, Devinder Singh Rana, Anil Kumar Bhalla, Anurag Gupta, Mohinder Pal Sachdeva, Rohit Singh Huirem, Kirti Chauhan, M. H. Yashavarddhan, Atul Basnal, Ritu Gupta, Prashant Kumar Mallick, Nirmal Kumar Ganguly

**Affiliations:** 1Department of Biotechnology and Research, Sir Ganga Ram Hospital, New Delhi 110060, India; naveen20kr@gmail.com (N.K.); huiremrohit14@gmail.com (R.S.H.); chauhankirti218@gmail.com (K.C.); yashvarddhan@gmail.com (M.H.Y.);; 2Department of Nephrology, Sir Ganga Ram Hospital, New Delhi 110060, Indiadrbhallaak@gmail.com (A.K.B.);; 3Department of Anthropology, University of Delhi, New Delhi 110007, India; architatandon6@gmail.com (A.T.); mpsachdeva@rediffmail.com (M.P.S.); 4Laboratory Oncology, Dr BRAIRCH, All India Institute of Medical Sciences, New Delhi 110029, India; atulbasnal2210@gmail.com (A.B.);; 5ICMR-National Institute of Malaria Research, Dwarka, Sector-8, New Delhi 110077, India

**Keywords:** dd-cfDNA, kidney transplant, minimally invasive biomarkers, rejection

## Abstract

Monitoring graft health and detecting graft rejection is crucial for the success of post-transplantation outcomes. In Western countries, the use of donor-derived cell-free DNA (dd-cfDNA) has gained widespread recognition as a diagnostic tool for kidney transplant recipients. However, the role of dd-cfDNA among the Indian population remains unexplored. The recipients were categorized into two groups: the post-transplant recipient (PTR) group (*n* = 16) and the random recipient (RR) group (*n* = 87). Blood samples were collected daily from the PTR group over a 7-day period, whereas the RR group’s samples were obtained at varying intervals. In this study, we used a targeted approach to identify dd-cfDNA, which eliminated the need for genotyping, and is based on the minor allele frequency of SNP assays. In the PTR group, elevated dd-cfDNA% levels were observed immediately after transplantation, but returned to normal levels within five days. Within the RR group, heightened serum creatinine levels were directly proportional to increased dd-cfDNA%. Sixteen recipients were advised to undergo biopsy due to elevated serum creatinine and other pathological markers. Among these sixteen recipients, six experienced antibody-mediated rejection (ABMR), two exhibited graft dysfunctions, two had active graft injury, and six (37.5%) recipients showed no rejection (NR). In cases of biopsy-proven ABMR and NR, recipients displayed a mean ± SD dd-cfDNA% of 2.80 ± 1.77 and 0.30 ± 0.35, respectively. This study found that the selected SNP assays exhibit a high proficiency in identifying donor DNA. This study also supports the use of dd-cfDNA as a routine diagnostic test for kidney transplant recipients, along with biopsies and serum creatinine, to attain better graft monitoring.

## 1. Introduction

Kidney transplant recipients (KTRs) enjoy favorable allograft function outcomes during the first year following transplantation. Substantial advancements have been achieved in mitigating acute T-cell mediated rejection; however, antibody-mediated rejection remains a notable concern, particularly in the later phases post-transplantation [1,2].

Currently, a deficiency exists in precise biomarkers capable of accurately gauging the extent of active injury or graft dysfunction in kidney transplantation. The evaluation predominantly relies on identifying functional injury indicators, including elevated serum creatinine levels, therapeutic drug monitoring, and screening for detrimental donor-specific antibodies. Of these, serum creatinine (sCr) stands as the most widely employed marker for diagnosing graft function in KTRs. Elevated serum creatinine levels manifest when there is a substantial reduction in glomerular filtration rate (GFR), signifying an approximate loss of 50% in kidney function [1,3]. Consequently, sCr is considered a late indicator in assessing graft rejection.

The current “gold standard” for evaluating graft status involves biopsy, a procedure fraught with inherent limitations such as invasive nature, prolonged turnaround times, interobserver variability, and high costs [1,4]. In renal transplant protocol biopsies, the incidence of hematuria and major complications has been documented at 3.5% and 1%, respectively [5]. Roughly one-quarter of these biopsies yield insufficient specimens, a rate that tends to be higher with smaller needle sizes [6]. Moreover, studies reveal that 73% of clinically indicated biopsies, based on elevated serum creatinine levels, do not uncover active rejection [7]. Correspondingly, another study reports that 43% of clinically indicated biopsies and 65% of protocol biopsies are deemed unnecessary invasive procedures [8]. Biopsies are not conducted routinely, but are triggered by suspicions of graft damage and rejection, often guided by serum creatinine levels and other pathological factors. Consequently, by the time a biopsy is undertaken, over 50% of the graft is already compromised [1,3,9,10].

The utilization of donor-derived cell-free DNA (dd-cfDNA) or graft cell-free DNA as a noninvasive biomarker for routine graft health monitoring and graft rejection diagnosis in kidney transplant recipients has been endorsed and embraced in western nations [7,11,12]. The prosperity and survival of the graft in kidney transplant recipients hinge on vigilant graft health monitoring, and the early identification of graft damage and rejection.

Detecting minute quantities of dd-cfDNA or donor DNA in the recipient’s plasma sample poses a challenge, yet it is attainable through advanced molecular biology methods. Various techniques exist for identifying and quantifying donor-derived cell-free DNA, including: (i) sequencing both donor and recipient genomes and pinpointing distinct single nucleotide polymorphisms (SNPs), (ii) recognizing the presence of the Y chromosome from a male graft in a female recipient, (iii) employing next-generation sequencing assays that use a panel of SNPs to discern donor DNA sans the requirement of a donor sample and sequencing, and (iv) employing a targeted approach where SNP assays are meticulously chosen based on minor allele frequency, and subsequently evaluated for applicability [1,4,13].

Despite the proven efficacy of dd-cfDNA, no investigations have delved into its role within the Indian population. This study endeavors to explore the viability of selected SNP assays among Indian KTRs. Employing a targeted methodology, this study quantifies dd-cfDNA in kidney transplant recipients using preselected SNP assays grounded in minor allele frequency (MAF), offering a means to monitor graft health and diagnose rejection without prior knowledge of donor genotypes.

## 2. Materials and Methods

This study was conducted solely at Sir Ganga Ram Hospital in New Delhi, India. It is a noninterventional study focusing on assessing the practicality and clinical significance of dd-cfDNA in identifying graft damage and rejection in KTRs. Graft injury was discerned through the examination of patients’ laboratory outcomes. Data collection took place between June 2021 and January 2022, during which time the patients received treatment, care, and immunosuppression regimens tailored to the discretion of attending physicians and surgeons. It is noteworthy that this study exerted no influence on patient treatment, monitoring, diagnosis, or medication administration.

Two distinct patient groups were established: (i) the post-transplant recipients’ group (PTR) and (ii) the random recipients’ group (RR). The PTR group encompassed patients who were enrolled from postoperative day 1 (POD1) to POD7 following the transplantation surgery. Daily blood samples were drawn from these individuals to track changes in dd-cfDNA%. Blood samples were collected daily from each of these patients for 7 days. In contrast, the RR group consisted of patients enrolled at random times during routine hospital visits for consultation. Blood samples were collected from these patients only once. Further, the RR group also included patients from PTR and the values of dd-cfDNA% obtained on POD7 was considered. This study protocol’s visual representation is delineated in Figure 1.

### 2.1. Inclusion and Exclusion Criteria

#### 2.1.1. Inclusion Criteria

Kidney transplant recipients who agreed to participate in the study and provided written consent.Recipients of a single organ and their first graft.

#### 2.1.2. Exclusion Criteria

Recipients with prior kidney grafts.Recipients of multiple organ transplants.Pregnant women.Individuals who have an active coronavirus disease 2019 (COVID-19) infection or any other infection.Individuals who are less than 1-month postdiagnosis of COVID-19 infection.

The rationale for excluding COVID-19 infected patients from this study was the lack of evidence regarding the association between cell-free DNA and COVID-19. In addition, the pandemic era witnessed significant modifications in immunosuppressive regimens, possibly resulting in elevated levels of donor-derived cell-free DNA in kidney transplant recipients.

### 2.2. Blood Collection and Storage

Around 5 mL of blood was collected using ethylenediaminetetraacetic acid tubes. Within an hour of collecting the samples, plasma and buffy coat components were separated. The blood sample was centrifuged at 2500× *g* for 10 min at 4 °C. The separated plasma and buffy coat were then preserved at −20 °C until subsequent processing.

### 2.3. Selection of SNP Assays

Beck and colleagues have previously identified 41 SNP assays based on an MAF of >43% across diverse reported ethnicities [1]. These assays were chosen considering their adherence to the Hardy–Weinberg equilibrium, facilitating differentiation between two individuals, irrespective of their ethnicity, caste, or relationship. Following the principles of the Hardy–Weinberg equilibrium, an SNP with an MAF ranging from 0.4 to 0.5 has an 11.5% to 12.5% probability of exhibiting distinct alleles between two individuals, which holds significance in transplantation, distinguishing the donor and recipient. These probabilities and percentages have been calculated using the established model for exclusion probabilities [1].

Within our study, we selected five SNP assays from the 41 proposed by Beck et al. [1]. This selection was based on the MAF specific to the Indian population, prioritizing SNPs with the highest MAF closest to 0.49, in a descending order. The distinct characteristics of the chosen SNPs are comprehensively outlined in Table 1. The allele frequencies, data from the National Library of Medicine of the National Institute of Health and IndiGenome were employed. A comprehensive account of MAF of the selected SNPs are shown in Table 2. For each SNP, a dedicated set of primers and probes was meticulously designed. The primer–probe sequences for these chosen assays are given in Appendix A.

### 2.4. Genomic DNA and Cell-Free DNA Extraction

To extract genomic DNA (gDNA) from the stored buffy coat, we used the Qiagen DNeasy Blood and Tissue Kit (Catalog number 69504). The manual was followed with slight modification in spin duration and temperature to enhance the purity and concentration. The elution volume for gDNA was maintained at 100 μL.

To extract cell-free DNA (cfDNA), we processed 1.5 mL of stored plasma using the Qiagen QIAmp Circulating Nucleic Acid Kit (Catalog number 55114). Before cfDNA extraction, the stored plasma underwent an additional centrifugation step at 4000× *g* for 20 min at 4 °C to eliminate any residual cell debris. The manual was followed with slight modification in spin duration and temperature to enhance the purity and concentration. The final elution volume for cfDNA was established at 60 μL.

### 2.5. Preamplification of cfDNA

The usual yield from 1 mL of EDTA plasma typically ranges from 2000 to 3000 genomic copies. Thus, the yield from 1.5 mL of EDTA plasma would fall within the range of 3000 to 4500 copies. Assuming that donor-derived cell-free DNA (dd-cfDNA) comprises 2% of the entire cell-free DNA, the estimated count of dd-cfDNA copies would be approximately 60 to 90 copies per 1.5 mL of plasma. Consequently, to accommodate testing for multiple SNPs, a preliminary amplification of cell-free DNA is essential [1].

To perform the preamplification of cell-free DNA, we used the Takara ThruPLEX DNA-Seq HV kit (Catalog number R400740). The preamplification procedure consisted of three stages: template preparation (with cycling conditions set at 22 °C for 25 min, 55 °C for 20 min, and a final step at 4 °C for 20 min), followed by library synthesis (cycling conditions: 30 °C for 40 min and 4 °C for 20 min), and library amplification (cycling conditions: 72 °C for 3 min, 85 °C for 2 min, 98 °C for 2 min, succeeded by 12 cycles of 98 °C for 20 s, 68 °C for 75 s, and a final extension at 68 °C for 5 min). The protocol furnished in the kit manual was meticulously adhered to.

Subsequent to the cell-free DNA preamplification, a PCR cleanup step was executed to eliminate superfluous reagents, dyes, and primers from the preamplification mixture, thereby yielding pristine cell-free DNA. The purification process engaged the Takara NucleoMag NGS Clean-up and Size Select kit (Catalog number 744970.5), in accordance with the outlined procedure in the kit manual, albeit with slight adjustments in incubation time to attain heightened levels of cell-free DNA purity and concentration. This method is based on the application of magnetic beads [2].

### 2.6. Recipient Genotyping

Recipient genotyping was executed using gDNA, employing the real-time PCR displacing probes technique. This approach involved the using of a pair of probes for each allele, leading to a fluorophore signal emission contingent upon the presence of the specific allele. For this genotyping procedure, the Roche FastStart^TM^ Taq DNA Polymerase, dNTPack kit (Catalog number 4738381001) was used.

Each genotyping reaction was composed of 100 ng of gDNA as the template, 2 μL of FastStart 10× buffer with MgCl_2_, 200 μM of dNTPs, 900 nM of each primer, 250 nM of each probe, 2 units of FastStart Taq, and, if needed, 1.8 μL of 25 mM MgCl_2_, as specified in Appendix A. The PCR conditions encompassed an initial denaturation stage at 95 °C for 10 min, succeeded by 50 cycles of denaturation at 95 °C for 30 s, and annealing/extension at 65 °C for 1 min.

### 2.7. dd-cfDNA Identification and Quantification

Utilizing the Qiagen QIAcuity digital PCR (dPCR) system, all reactions were prepared using the Qiagen QIAcuity Probe PCR kit (Catalog number 250102). Each reaction incorporated 100 ng of preamplified cfDNA as the template, along with 800 nM of each primer and 400 nM of each probe. The PCR reaction mixture was distributed among approximately 26,000 partitions, in accordance with the specifications of the Qiagen QIAcuity Nanoplate 26 k 24-well (Catalog number 250001). The PCR conditions were set as 95 °C for 3 min, followed by 40 cycles of denaturation at 95 °C for 15 s, and annealing/extension at 65 °C for 40 s.

Following the termination of the PCR cycle, each partition was scrutinized for the presence or absence of the targeted sequence. The comprehensive master mix reaction of 40 μL was divided into approximately 26,000 sections, with the fluorescence being individually detected within each partition.

## 3. Results

A total of 117 kidney transplant patients underwent initial assessment and were subsequently categorized into two distinct groups: the PTR group and the RR group, comprising 18 and 99 patients, respectively, as illustrated in Figure 2.

Within the PTR group, two patients withdrew their consent for participation, leading to their exclusion from this study. Correspondingly, within the RR group, twelve patients were excluded due to their COVID-19 infection status or their history of prior or multiple grafts. The final analysis consisted of 103 patients, forming the basis for evaluating the utility and applicability of the selected SNPs. Upon scrutinizing the chosen SNPs to identify informative assays, it was ascertained that one patient from the PTR group and six patients from the RR group lacked informative assays. Consequently, these individuals were excluded from subsequent analyses.

The characteristics and attributes of the analyzed patient cohort are comprehensively presented in Table 3. The patients exhibited a mean age of 39.5 ± 12.0 years, with males constituting 81% of the participant pool. Hypertension was determined as the predominant cause of kidney failure, representing 62.5% of the patients and serving as the primary indication for kidney transplantation. The etiologies observed for kidney failure among the study participants encompassed a range of factors, including diabetes, kidney atrophy, polycystic kidney disease (PCKD), nephrotic syndrome, lupus nephritis, focal segmental glomerulosclerosis (FSGS), calcification in renal medullary region (CRMR), immunoglobulin nephropathy (IGNF), excessive painkiller consumption, and unknown reasons.

### 3.1. Performance of the Selected SNP Assays

Each patient’s gDNA underwent RT-PCR, while cfDNA was subjected to dPCR for the identification of informative assays. During the RT-PCR phase, if an SNP assay exhibited a heterozygous genotype in the recipient, it was excluded from further investigation, as its accurate quantification using dPCR could not be ensured. The subsequent identification process was executed on the amplified cell-free DNA through dPCR. An assay was deemed informative when the recipient displayed a homozygous genotype while the graft exhibited the heterologous allele, in either a heterozygous or homozygous state.

Upon employing the selected SNP assays based on Indian MAF, it was observed that among the 103 recipients, 7 (6.80%) individuals did not have any informative assays, yielding an efficiency rate of 93.2% for the selected SNPs in detecting donor-derived cell-free DNA in our study. Among the recipients, the majority, 36 (34.95%) exhibited a single informative SNP assay, followed by 26 (25.24%) recipients with two informative assays. Additionally, 21 (20.39%) recipients showcased three informative assays, while 9 (8.74%) and 4 (3.88%) recipients demonstrated four and five informative assays, respectively, as represented in Figure 3.

### 3.2. Post-Transplant Recipient Group

The PTR group consisted of 16 patients. Blood samples were collected daily for 7 days from these patients for 7 days post-transplantation. This will help in the analysis of the concentration pattern and daily fluctuations in dd-cfDNA levels in the first week of transplant. Among the 16 patients, one was excluded from the analysis due to the absence of informative SNPs among the selected assays. As a result, a total of 90 samples from 15 patients within the PTR group were subjected to analysis. The dd-cfDNA% values for each patient over the course of seven days are presented in Figure 4.

It is evident that approximately 5 days after transplantation, the median dd-cfDNA% values returned to the normal range. The median (IQR) dd-cfDNA% values for each post-operative day (POD) were as follows: 3.21 (2.28–4.88) on POD1, 1.41 (1.32–2.10) on POD2, 0.83 (0.57–0.97) on POD3, 0.42 (0.38–0.71) on POD4, 0.31 (0.18–0.48) on POD5, 0.29 (0.20–0.43) on POD6, and 0.25 (0.13–0.45) on POD7, as illustrated in Figure 5.

### 3.3. Random Recipient Group

A comprehensive analysis encompassed a total of 87 patients within the RR group. Six patients were excluded from further analysis due to the absence of informative SNP assays. Notably, the RR group also incorporated 15 patients from the PTR group, wherein the dd-cfDNA value on the 7th day post transplantation was considered for analysis. Thus, a cumulative total of 96 patients, along with their corresponding samples, were subject to thorough examination. The distribution of dd-cfDNA% among these 96 patients is visually depicted in Figure 6.

Within the RR group, patients were classified into three distinct categories, based on the percentage of dd-cfDNA present in the recipients’ plasma. These categories comprised: stable (<0.5%), likely indicating active injury and/or rejection (0.5–1.0%), and posing a high risk of rejection (>1%). The median (with interquartile range) serum creatinine levels for each category were calculated and graphically depicted, as illustrated in Figure 7. The stable group exhibited a median serum creatinine level of 1.21 (0.95–1.28) mg/dL, the category at likely risk of active injury had an average serum creatinine level of 1.63 (1.27–2.01) mg/dL, and finally, the high risk of rejection group displayed an average serum creatinine level of 1.93 (1.22–2.36) mg/dL.

Among these recipients, 16 recipients underwent graft biopsy due to clinical indications involving serum creatinine levels and other pathological tests. Of these recipients, six (37.5%) were diagnosed with antibody-mediated rejection (ABMR), exhibiting a mean ± SD dd-cfDNA% of 2.80 ± 1.77 and a median (IQR) dd-cfDNA% of 2.44 (1.45–3.91). Additionally, two (12.5%) recipients displayed active graft injury, presenting a mean ± SD dd-cfDNA% of 0.88 ± 0.09, while another two (12.5%) exhibited graft dysfunctions (GD) with a mean ± SD dd-cfDNA% of 0.54 ± 0.01. Six (37.5%) recipients who had clinically recommended biopsies exhibited no rejection status. Within this subgroup, the mean ± SD dd-cfDNA% was 0.30 ± 0.35, and the median (IQR) dd-cfDNA% was 0.13 (0.09–0.34).

## 4. Discussion

Challenges to graft longevity encompass two primary factors: irreversible chronic rejection and the adverse effects associated with standard immunosuppression. These factors contribute to a spectrum of complications, including nephrotoxicity, cardiovascular ailments, susceptibility to opportunistic infections, and an elevated risk of malignancy [1,2]. Despite notable enhancements in the one-year survival rate of transplanted kidneys, achieving long-term graft survival and effective management necessitates further advancements [14,15,16].

The currently used diagnostic tests for kidney rejection, such as serum creatinine, manifest abnormalities when graft damage has already reached or exceeded 50%. Similarly, graft biopsy, an invasive procedure, is not conducted on a regular basis. Instead, it is performed when any functional irregularities are suspected, often based on serum creatinine levels and other pathological markers. Consequently, both serum creatinine and biopsy serve as late markers, leading to delayed interventions, graft impairment, and potential rejection. This highlights the need for a novel biomarker that possesses attributes of accessibility, noninvasiveness, reproducibility, and heightened sensitivity and specificity [1,3,4,13]. The detection of donor or graft DNA in recipients’ plasma holds promise as a biomarker for solid organ graft rejection due to remarkable sensitivity and specificity. Clinical trials have validated kidney graft monitoring and rejection diagnosis through dd-cfDNA. However, standardizing diagnostic thresholds and performance parameters remains a challenge, with some discrepancies emerging across studies [1,2,11,17,18,19,20]. While many studies focus on the relative fraction of donor derived cell-free DNA (dd-cfDNA%), a growing body of evidence supports the superiority of absolute quantification of donor cell-free DNA (dd-cfDNA copies/mL) for graft monitoring due to its dependency on various physiological conditions [21]. Dd-cfDNA can distinguish among rejection and stable cases efficiently, but when it comes to distinguishing between antibody-mediated rejection and T-cell-mediated rejection, there are still gaps in the knowledge. Few studies reported to distinguish the type of rejection based on the dd-cfDNA values, whereas, some refused to find any significant difference. Similarly, dd-cfDNA cannot distinguish among the different forms of graft dysfunction [11,19].

Beck et al. established a combination of assays enabling the identification and quantification of dd-cfDNA [1,20]. This study represents the first from India to assess the role of dd-cfDNA in kidney transplantation. In this study, we investigated the applicability and efficiency of five selected SNPs in the Indian population. Utilizing these SNPs, our initial objective was to distinguish and identify donor DNA from the recipient’s plasma without the need for genotyping. The selection of these SNPs was guided by the MAF in the Indian population. Among the 103 patients, seven (6.8%) exhibited noninformative SNPs. Notably, 34.95% of patients had one informative SNP, while 25.24% exhibited two informative SNPs out of the selected five SNP assays. All five informative assays were observed in 3.88% of recipients. Our study employed a smaller number of SNP assays compared to others, which often involve a more extensive assessment to identify informative SNPs [1,2,7,12,19,20,21,22]. The unique contribution of our study lies in the selection of SNP assays that had not been previously tested within the Indian population.

Donor-derived cell-free DNA represents an independent biomarker unaffected by bodily functions, physiology, or conditions. Its concentration correlates directly with the number of deteriorating graft cells. Hence, higher donor DNA concentrations indicate an elevated risk of graft damage and subsequent rejection [23,24]. Consequently, early detection of acute or cellular rejection becomes feasible, and unnecessary biopsies can be avoided. Elevated dd-cfDNA concentrations during the first post-transplant week are presumably attributed to ischemia and reperfusion injury. This pattern is also evident in our PTR group, where 15 patients were monitored over a seven-day period. Concentrations of dd-cfDNA% gradually decrease and fall below the threshold value after POD5.

Our comprehensive analysis encompassed a cohort of 96 patients, of which 15 were from the PTR group. In our study, recipients were categorized into three groups based on dd-cfDNA concentration for analysis, revealing dd-cfDNA% in relation to serum creatinine levels. Stable patients maintained serum creatinine levels within the normal range, while those at risk of graft damage and rejection demonstrated elevated levels. Conversely, patients categorized as high-risk based on dd-cfDNA% exhibited significantly elevated serum creatinine levels. Furthermore, sixteen recipients underwent clinically indicated biopsy, and the mean ± SD dd-cfDNA% for biopsy-proven ABMR was 2.80 ± 1.77, surpassing the proposed threshold values of graft rejection (>1%) for KTRs. Additionally, the mean ± SD dd-cfDNA% for biopsy-proven active injury and graft dysfunction were 0.88 ± 0.09 and 0.54 ± 0.01, respectively, placing them within the active injury category (dd-cfDNA% of 0.5–1.0). Lastly, recipients who showed no evidence of rejection in the biopsy, but had elevated serum creatinine, exhibited a mean ± SD dd-cfDNA% of 0.30 ± 0.35, with a median (IQR) dd-cfDNA% of 0.13 (0.09–0.34). These biopsies are an unnecessary invasive procedure that might be avoided by implementing dd-cfDNA tests in regular monitoring of the kidney transplant recipients. These results align with international studies on KTRs [1,2,11,12,18,19,21,22]. This indicated that serum creatinine is less sensitive and specific to diagnose rejection, and an elevation in serum creatinine is not always associated with graft dysfunction or rejection. This study indicated the use of new markers for graft monitoring so that the graft survival time can be increased.

Tailoring immunosuppressive regimens to transplant patients is critical due to the enduring need for lifelong immunosuppression. Striking the optimal balance between suboptimal dosing leading to allograft rejection and the potential hazards associated with excessive dosing, including nephrotoxicity and other complications, is paramount [2]. A promising avenue for achieving personalized immunosuppressive therapy lies in dd-cfDNA, offering the potential to extend graft lifespan and recipient well-being.

In India, although cell-free DNA-based tests have recently been introduced, their widespread acceptance and adoption as routine diagnostic tools for monitoring remain limited. The prohibitive cost of these tests stands as a primary barrier. Thus, it is imperative to devise strategies for developing more affordable testing solutions without compromising accuracy or reliability. According to this study, the results of donor-derived cell-free DNA-based diagnosis can provide results in less than 24 h, and would cost around USD 150 for first-time patients, and <USD 100 for repeat patients, which is much less expensive than a tissue biopsy.

## 5. Conclusions

Our study suggests that the selected SNP assays may exhibit high efficacy among the Indian population, potentially contributing to a reduction in the cost of testing for recipients. We observed that recipients with biopsy-proven rejection had dd-cfDNA concentrations exceeding 1%, which aligns with findings from other studies conducted worldwide. Additionally, we noted that elevated dd-cfDNA levels in the early post-transplant period gradually decreased and reached values below the specified threshold within approximately 5 days post-transplant.

It is important to note that this study was limited in scope, focusing on a small number of preselected SNPs. Further, this study accounts for only 16 biopsy-proven results, which is small in scope. It is important to evaluate the performance in a larger and more diverse population to ensure their reliability and generalizability across different transplant settings. By conducting further research, we can strengthen the evidence base and potentially expand the use of these assays in clinical practice for improved monitoring and management of transplant recipients.

## Figures and Tables

**Figure 1 diagnostics-13-03540-f001:**
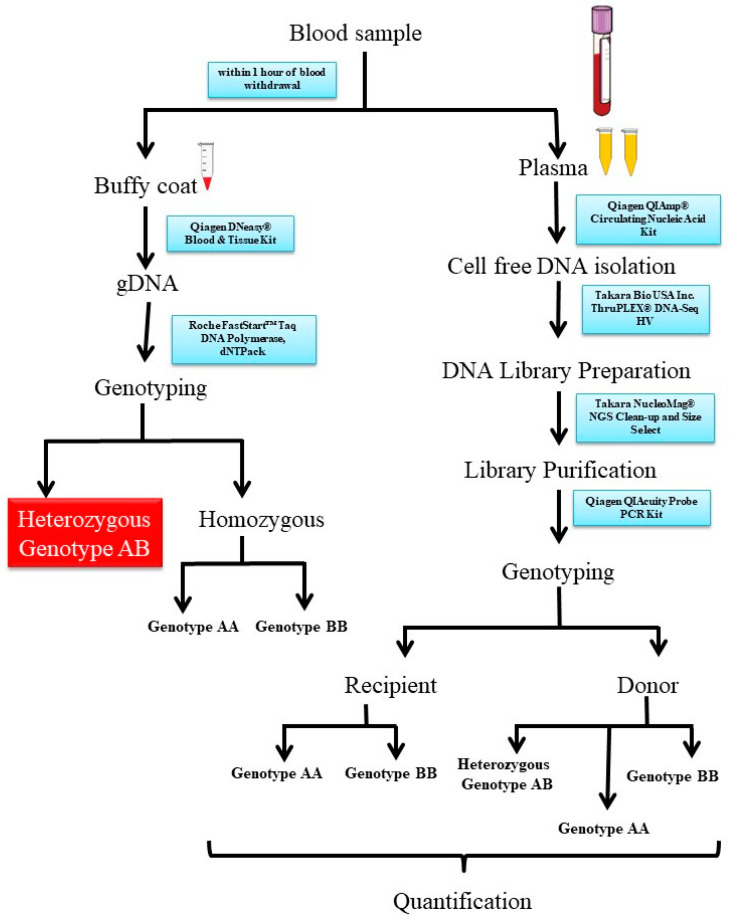
Schematic diagram of the methodology for quantification of dd-cfDNA.

**Figure 2 diagnostics-13-03540-f002:**
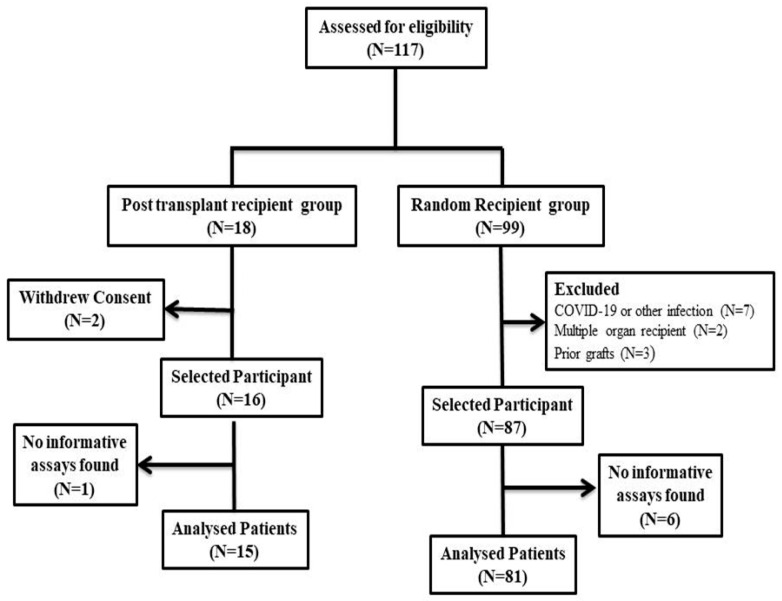
Selection of patients with predetermined inclusion criteria.

**Figure 3 diagnostics-13-03540-f003:**
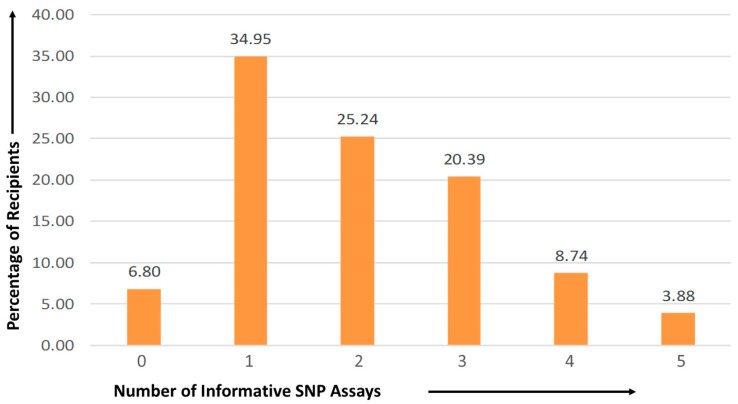
Distribution of informative SNP assays found among kidney transplant recipients.

**Figure 4 diagnostics-13-03540-f004:**
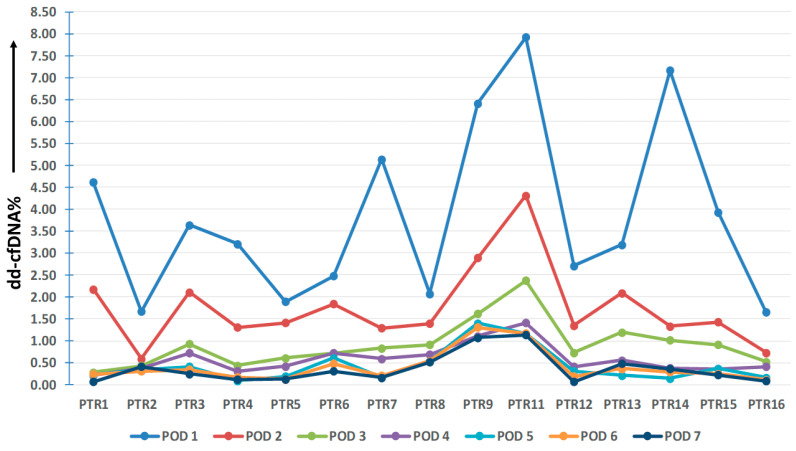
Time-course in plasma dd-cfDNA% during the first week of transplant.

**Figure 5 diagnostics-13-03540-f005:**
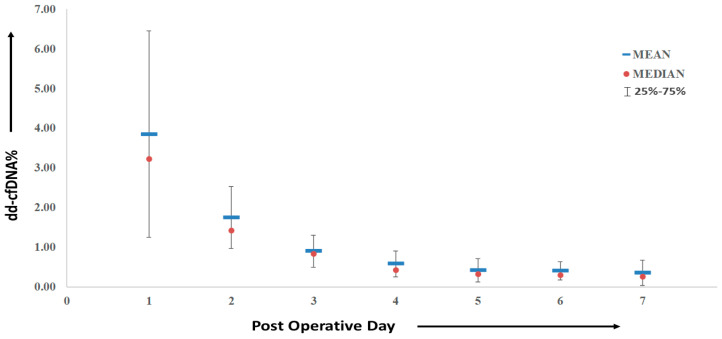
Time-course in dd-cfDNA% during the first week after kidney transplant recipients.

**Figure 6 diagnostics-13-03540-f006:**
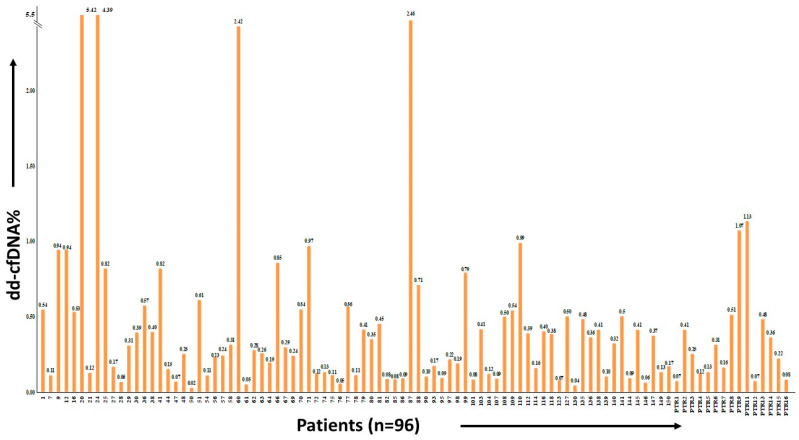
dd-cfDNA% in the reference population (*n* = 96).

**Figure 7 diagnostics-13-03540-f007:**
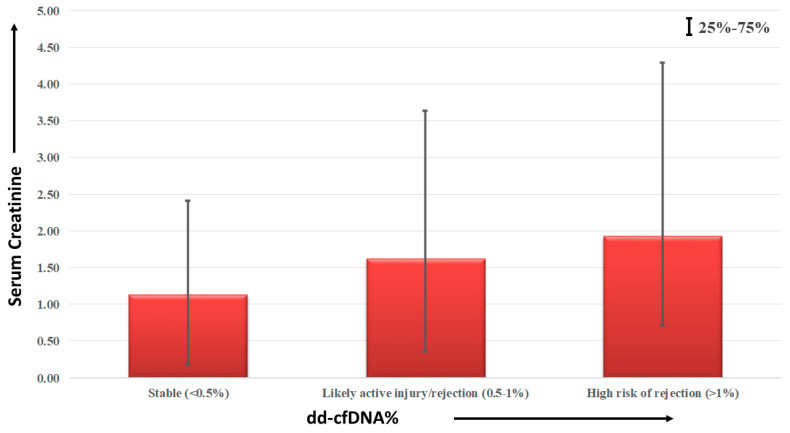
dd-cfDNA% vs. serum creatinine among the PTR group.

**Table 1 diagnostics-13-03540-t001:** Characteristics of the selected assays.

Assay Name	SNPs	Chromosome	Position	SNP	Length of Amplicon (bp)
S82	rs10228737	7	4,198,445	T > C	79
S87	rs10734083	10	129,154,730	T > C	94
S99	rs12064796	1	19,815,427	A > G	96
S103	rs4632826	5	142,485,948	T > C	143
S108	rs11610836	12	112,765,162	T > C	96

**Table 2 diagnostics-13-03540-t002:** Minor allele frequency of the selected assays.

Assay Name	SNPs	Global MAF	IndiGenome MAF	South Asian MAF	1000Genome South Asian MAF	HapMap South Asian MAF
S82	rs10228737	46.6%	46.8%	44.8%	43.4%	26.4%
S87	rs10734083	47.7%	45.2%	45.8%	44.4%	41.5%
S99	rs12064796	49.5%	49.7%	47.9%	48.4%	40.9%
S103	rs4632826	49.8%	47.9%	49.0%	48.5%	43.7%
S108	rs11610836	40.2%	49.1%	48.7%	46.9%	38.1%

**Table 3 diagnostics-13-03540-t003:** Characteristic features of the study population.

Characteristic	Frequency (*n* = 96)	Percent or Mean ± SD
Age (years)	96	39.5 ± 12.0
Sex		
Male	78	81.20%
Female	18	18.80%
Indications for kidney transplant		
CRMR	1	1.00%
Diabetes	8	8.30%
FSGS	2	2.10%
Hypertension	60	62.50%
IGNF Nephropathy	1	1.00%
Kidney Atrophy	6	6.30%
Lupus Nephritis	2	2.10%
Nephrotic Syndrome	3	3.10%
Painkillers	3	3.10%
PCKD	5	5.20%
Unknown	5	5.20%
Donor Type *		
Deceased	2	2.10%
Living	94	97.90%
ABO compatible		
Compatible	83	86.50%
Incompatible	13	13.50%

* Deceased donors, *n* = 2 (1.9%) are unrelated and unknown donors. Whereas, living donors comprised—brother—6 (5.8%); cousin—4 (3.9%); daughter—2 (1.9%); family friend (unrelated)—1 (1.0%); father—7 (6.8%); grandmother—1 (1.0%); husband—2 (1.9%); in-laws—5 (4.9%); maternal aunt—3 (2.9%); maternal uncle—2 (1.9%); mother—33 (32.0%); paternal aunt—2 (1.9%); paternal uncle—1 (1.0%); sister—5 (4.9%); son—1 (1.0%); step mother—1 (1.0%); unrelated—1 (1.0%); wife—24 (23.3%).

## Data Availability

The data will be made available on reasonable request from the corresponding author.

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
