# Peer review of "Donor-Derived Cell-Free DNA as a Non-Invasive Biomarker for Graft Rejection in Kidney Transplant Recipients: A Prospective Study among the Indian Population"

_diagnostics, 2023, doi:10.3390/diagnostics13233540_

Round 1

Reviewer 1 Report

The aim of this prospective study was to apply the already established biomarker “donor-derived cell-free DNA” to an Indian population of 103 kidney transplant patients. The author’s conclusion is the “support of the use of dd-cfDNA as a routine diagnostic”.

Here my points of criticism:

Line 17:  A connecting phrase between general remarks on cell-free DNA and the study description would be helpful.

Line 48: “sampling errors”. This can also happen with cell-free DNA samples. Overall, the manuscript contains many general meaningless statements on general aspects of quality assurance, which should be avoided.

Line 60: What is the difference between “donor-derived cell-free DNA” and “graft cell-free DNA”. Please explain.

Line 90, 91: Please distinguish the two patient groups in detail. For me, their differences remain unclear.

Line 109: Why COVID-patients have been excluded from the study?

Line 119: Reference 14 corresponds to https://cen.acs.org/articles/96/i5/uncovering-the-hidden-signs-of-organ-transplant-rejection.html, which is not by „Beck et al.”.

 Line 129: See comment above

Line 143 – 155: This paragraph as others are much lo wordy and contains superfluous information (“the manual was followed…”).

 Line 196:  See comment above

Line 204: “Results” rather than “Result”

Line 250 and line 269: Again, differentiate the two patient groups in detail.

Line 200 – 306: Again, very general textbook information not leading to the goal the study.

Line 305: Reference (1) is now “Beck et al. cited above.

General remarks on the manuscript: Why do the author’s describe long –term complications in the context of a follow-up period of seven days. Does it make sense to state the “remarkable sensitivity and specificity” in a paper which wants to explore especially this topic?

Line 326: Again, Ref 14 is something different (C&EN, impact factor 0,3).

Line 343: How can you related dd-cfDNA concentrations decisively to ischemia and reperfusion and not to rejection? Please explain.

Author Response

Reviewer 1

The aim of this prospective study was to apply the already established biomarker “donor-derived cell-free DNA” to an Indian population of (n=103) kidney transplant patients. The author’s conclusion is the “support of the use of dd-cfDNA as a routine diagnostic.”

Here my points of criticism:

(C) Line 17:  A connecting phrase between general remarks on cell-free DNA and the study description would be helpful.

The authors would like to thank you for the valuable insight and we have incorporated this suggestion.

(C) Line 48: “sampling errors”. This can also happen with cell-free DNA samples. Overall, the manuscript contains many general meaningless statements on general aspects of quality assurance, which should be avoided.

The authors would like to thank for pointing at this. And we have incorporated the changes and removed the “sampling error” from the manuscript.

 (C) Line 60: What is the difference between “donor-derived cell-free DNA” and “graft cell-free DNA”. Please explain.

Both graft cell-free DNA and donor-derived cell-free DNA are the same. Different authors have used these two terms for the same donor derived or graft derived cell free DNA in case of organ transplant recipients.

(C) Line 90, 91: Please distinguish the two patient groups in detail. For me, their differences remain unclear.

The authors thank you for the comments and we have rewritten the lines and we hope that this would be clear for the readers.

(C) Line 109: Why COVID-patients have been excluded from the study?

The association of COVID-19 disease and cell free DNA is unknown. As cell free DNA are released on the breakdown of the cells due to apoptosis and necrosis. We are not sure if it is associated with COVID-19 or not. The other reason to exclude COVID-19 infected patients is the changes in the immunosuppressive regimes during COVID-19 infection which may have led to increased donor derived cell free DNA.

I hope we have answered your query.

(C) Line 119: Reference 14 corresponds to https://cen.acs.org/articles/96/i5/uncovering-the-hidden-signs-of-organ-transplant-rejection.html, which is not by „Beck et al.”.

We have incorporated the change and the corrected the reference.

(C) Line 129: See comment above

We have incorporated the change and the corrected the reference.

(C) Line 143 – 155: This paragraph as others are much lo wordy and contains superfluous information (“the manual was followed…”).

We have rephrased the line and written in brief and crisp manner.

(C) Line 196:  See comment above

We have rephrased the line as per your suggestions.

(C) Line 204: “Results” rather than “Result”

We have incorporated the change and added ‘s’ in the Result.

(C) Line 250 and line 269: Again, differentiate the two patient groups in detail.

We have rephrased the line as per your suggestions to make it clearer and easier to understand.

(C) Line 200 – 306: Again, very general textbook information not leading to the goal the study.

Thank you for you for pointing out. I think you are intending to mention line 300-306.  These lines were written to give a general information (as mentioned by you). It can be deleted as suggested by you.  

(C) Line 305: Reference (1) is now “Beck et al. cited above.

Reference (1) is correct here as suggested.

(C) General remarks on the manuscript: Why do the authors describe long –term complications in the context of a follow-up period of seven days. Does it make sense to state the “remarkable sensitivity and specificity” in a paper which wants to explore especially this topic?

The term ‘long-term’ mentioned (line 306) refers to the long-term survival of graft to describe the need of better monitoring tool for assessing graft health.

The “remarkable sensitivity and specificity” written in the discussion section is written based on the global studies based on clinical trials and other research studies. It is also used as a routine marker along with biopsy and other pathological markers in western countries.

These both terms are written to describe the need and efficiency of dd-cfDNA and not based on this research.

Based on our research article We have written, “These results align with international studies on KTRs (1,2,11,12,20,21,23,24). This indicated that serum creatinine is less sensitive and specific to diagnose rejection and the elevations in serum creatinine is not always associated with graft dysfunction or rejection. This study indicated to the use of new markers for monitoring of graft so that the graft survival time can be increased.”

I hope I made myself clear to you and resolved the doubt.

(C) Line 326: Again, Ref 14 is something different (C&EN, impact factor 0,3).

The reference has been corrected.

(C) Line 343: How can you related dd-cfDNA concentrations decisively to ischemia and reperfusion and not to rejection? Please explain.

We appreciate the comment and we have incorporated the changes accordingly. It is presumed that it is because of ischemia/ reperfusion. We have incorporated changes accordingly and have added word presumably.

Reviewer 2 Report

This was a very well written article, although at times was hard to follow. The abstract would suggest that more patients were included, although only 15 post tx patients and another 81 random recipients were found to have an "informative assay". Many units routinely use protocol biopsies and this is both not referenced, but stated by the authors not to occur. It is interseting to note that dd cf DNA falls by day 5 post Tx, which is typically the time point that cell mediated rejection occurs. Hence this would appear to completely contradict your conclusion that dd cf DNA can be used as a biomarker in place of serum Cr or biopsy. No discussion about the turn around time for the assays, compared with histology, or the added costs involved? You do not state that in the longer term (Random recipient gp) dd cf DNA cannot differentiate between the different forms of graft dysfunction, although can differentiate between ABMR and no rejection after they have been confirmed with biopsy. One of the major problems is the very low incidence of biopsy being only 16 from the whole cohort. It is not clear whether the live donors (94 of the 96 patients) were related or unrelated and this may well impact both the assays, but also the incidence of acute rejection/graft dysfunction. I would also suggest excluding the 13 ABOi transplants as this further dilutes any comprehensible conclusions. Finally your conclusion that this work supports the use of dd cf DNA as a routine diagnostic tool is a very confident statement which as a reviewer I would have to disagree with from what you have found, but may become a useful adjunct with protocol biopsies and the routine use of biopsy in any case of graft dysfunction, alongside dd cf DNA to gain further knowledge and ability to interpret the levels alongside the current gold standard of biopsy.   Your use of biopsy is so low that it is very hard to accept validity of this assay into routine clinical practice.   

Author Response

The abstract would suggest that more patients were included, although only 15 post tx patients and another 81 random recipients were found to have an "informative assay".

(C) This was a very well written article, although at times was hard to follow.

The authors would like to thank you for your appreciation. We have made certain changes in the manuscript that will help in better understanding of the article.

(C) Many units routinely use protocol biopsies and this is both not referenced, but stated by the authors not to occur.

We have written “Biopsies are not conducted routinely but are triggered by suspicions of graft damage and rejection, often guided by serum creatinine levels and other pathological factors” which cited properly as reference number 1,3,9 and10 in the article.

(C) It is interesting to note that dd cf DNA falls by day 5 post Tx, which is typically the time point that cell mediated rejection occurs. Hence this would appear to completely contradict your conclusion that dd cf DNA can be used as a biomarker in place of serum Cr or biopsy.

The fall of dd-cfDNA by 5th day is presumed to be due to the reperfusion and ischemia. You have raised a very good point that cell mediated rejection occurs during this time.

We would like to clarify that, this is an exploratory study, where we have tested the applicability of the selected 5 markers in the Indian population and its utility as a diagnostic tool. As per the literature and previous studies, dd-cfDNA monitoring should be avoided for first 7 days post-transplant to exclude all these dilemmas of ischemia or cell mediated rejection. We have assessed the first seven days to track the changes and fluctuations as a part of an exploratory study.

This study does not conclude to replace the existing markers serum creatinine or biopsy. But the study supports that dd-cfDNA has higher potential than these two markers. dd-cfDNA can predict graft rejection with higher sensitivity, specificity than serum creatinine and can be helpful in avoiding unnecessary biopsies. Biopsies are essential to identify the type of rejection and it is not possible to replace biopsies at this point of time.

(C) No discussion about the turnaround time for the assays, compared with histology, or the added costs involved?

We have incorporated the changes and added it in the discussion.

(C) You do not state that in the longer term (Random recipient gp) dd cf DNA cannot differentiate between the different forms of graft dysfunction, although can differentiate between ABMR and no rejection after they have been confirmed with biopsy.

We have incorporated the suggested changes and is added in the manuscript.

(C) One of the major problems is the very low incidence of biopsy being only 16 from the whole cohort.

This is one of the limitations of this study. We are mentioning this in our manuscript as well based on your suggestion.

(C) It is not clear whether the live donors (94 of the 96 patients) were related or unrelated and this may well impact both the assays, but also the incidence of acute rejection/graft dysfunction.

We appreciate the suggestion and we have added the details of the donors in the manuscript.

(C) I would also suggest excluding the 13 ABOi transplants as this further dilutes any comprehensible conclusions.

This is a very valuable comment, and we would like to state that there are no studies where compatible or non-compatible transplants are discussed and this might not have much impact on the donor derived cell free DNA levels. Further I will consider this point in my future research and investigate the impact of ABOi transplants.

(C) Finally your conclusion that this work supports the use of dd cf DNA as a routine diagnostic tool is a very confident statement which as a reviewer I would have to disagree with from what you have found, but may become a useful adjunct with protocol biopsies and the routine use of biopsy in any case of graft dysfunction, alongside dd cf DNA to gain further knowledge and ability to interpret the levels alongside the current gold standard of biopsy.  

The statement to support dd-cfDNA as a routine diagnostic tool is in support of other studies which have similar results in kidney transplant recipients. The statement is not made based on this study. We have mentioned clearly in the conclusion section that a larger study is required for generalizability and further research is needed.

Further in the discussion section it is written that the results are aligning with other studies and it indicates to the use of new marker for monitoring.

Lastly, we have incorporated changes in the abstract to make it less strong and supporting the dd-cfDNA marker along with biopsies.

(C) Your use of biopsy is so low that it is very hard to accept validity of this assay into routine clinical practice.   

Thank you for your comment and we really appreciate this concern. The number of biopsies in this study is very low. And we are not making any statement based on these results in our study.

Being an exploratory study and the results being aligned and like the other studies worldwide, we are expressing our support. We have mentioned in the conclusion that further studies are required before making any generalized statement.

Round 2

Reviewer 1 Report

The authors have adequately responded to my points of criticism.

However: Please explain the exclusion of COVID-patients from the study not only to me but also shortly to the reader.

Author Response

The authors have adequately responded to my points of criticism.

(C) However: Please explain the exclusion of COVID-patients from the study not only to me but also shortly to the reader.

We appreciate the suggestions and as per your recommendations we have included the explanation in the manuscript highlighted in pink.